# Empirical Research on Early Internationalization of Firms in Sufficiently-Sized Domestic Market Country

**Saki Otomo [1,\*], Shuichi Ishida [1] and Mariko Yang-Yoshihara [1,2]**

[1] Department of Management Science and Technology, School of Engineering, Tohoku University, 6-6-11-803 Aza-Aoba Aramaki, Aoba-ku, Sendai 980-8579, Miyagi, Japan

[2] Freeman Spogli Institute for International Studies (FSI), Stanford University, 616 Jane Stanford Way, Encina Hall E005, Stanford, CA 94305-2004, USA

[\*] Correspondence: sakiootomo51@gmail.com

**Abstract:** Early internationalization and success in foreign markets play an important role in both a firm's growth and its impact on the global economy. We conducted a study on Japanese high-tech startups to investigate the factors that derive early internationalization in firms founded in countries with a large domestic market, despite the absence of strong incentives to operate overseas. Quantitative data were collected from 71 startups and analyzed with PLS-SEM (Partial least squares path modeling). Our result showed that the factors we extracted from the previous studies on the internationalization process in small-size markets would also apply in countries with large domestic markets. In addition, considerations and the types of technology, which we extracted from qualitative research, verified the effect. According to our mediator analysis, an entrepreneur's international orientation explains certain conditions related to a domestic market that affect a firms' decision to pursue early internationalization. Our study makes contributions at multiple levels, benefiting entrepreneurs who are considering overseas expansion as well as policymakers who aim to promote early internationalization efforts.

**Keywords:** internationalization; entrepreneurship; born global; innovation; startups; innovative organization

## 1. Introduction

Traditionally, firms are considered to take an incremental approach to their internationalization process. As the Uppsala model (Johanson and Vahlne 1977) predicts, firms are more likely to take on their internationalization efforts in markets with a greater degree of psychological proximity. Psychological distance refers to the differences between countries in terms of their language, culture, political systems, educational attainment, level of industrial development, etc. (Johanson and Wiedersheim-Paul 1975). This is due to the fact that entering a new and unfamiliar market involves inherent uncertainties and significant cost.

However, there are instances where young firms with limited resources and experience may choose to enter international markets soon after inception. These firms, typically small and technology-oriented, tend to adopt a globally-oriented vision from the outset and embark on rapid and dedicated internationalization through various modes of market entry, bypassing conventional stages of the internationalization process (Knight and Cavusgil 1996). They are commonly referred to as "Born Globals (BG, hereinafter)" (Rennie 1993) and "International New Ventures (INV, hereinafter)" (Oviatt and McDougall 1994).

This phenomenon has been most prevalent among knowledge-intensive firms including ones that specialize in software or information technology products that operate in small countries because the small domestic market drives them to seek opportunities beyond national borders and facilitate their entry into the international market (Lopez et al. 2009). Various researchers (McNaughton 2003; Fan and Phan 2007; Kudina et al. 2008, etc.) point out the size of the domestic demand as an important factor explaining a firm's early

internationalization. Rennie (1993), who coined the term BG, maintains that firms with smaller domestic markets are likely to internationalize earlier and faster through economies of scale and the pursuit of profit.

How, then, do firms with large a domestic market achieve early internationalization, despite the absence of strong incentives to operate overseas? Kudina et al. (2008) investigated the emergence of international new ventures among high-tech firms in the UK. The authors conclude that the drivers behind this phenomenon differ for firms originating in large markets (such as the United States or Japan), medium-sized markets (such as the United Kingdom), and small markets (such as New Zealand). A firm's early internationalization and success in foreign markets play an important role in promoting global economy and innovation (Oviatt and McDougall 1994), and therefore, firms with significant domestic markets should contemplate early internationalization as a key component of their strategic planning.

Otomo et al. (2023) conducted a study to identify the factors that derive early internationalization in firms founded in a country with a large domestic market, Japan.

Data were collected from 13 high-tech Japanese firms and classified based on the timing of their internationalization. We found that, in most cases, firms that proactively engaged in global markets from their inception were driven by the entrepreneur's strong international orientation. We also found that, for firms that entered the international market shortly after their inception, despite the absence of a clear vision of operating overseas, the condition of their originating market played a significant role in their decisions. This study identified additional factors, such as geographic considerations, global needs, and the type of technology, that may have influenced a firm's decision to enter the international market. This study suggests that many of the factors that facilitate internationalization, as previously identified in small domestic markets, may also apply in countries with larger domestic markets.

This study quantitatively tests the effect of the factors identified in the previous study and investigates what derives the high-tech startups in a large domestic market to achieve early internationalization despite the absence of strong incentive to operate overseas.

Our study will make contributions at multiple levels, benefiting entrepreneurs who are contemplating international operations in the future by helping to generate hypotheses that can serve as guiding posts for firms considering overseas operations as well as policymakers who aim to promote early internationalization efforts by providing insight into what external factors are asked to promote firms international business.

Theoretically, our study will expand the existing discussion on internationalization processes. Specifically, we will examine the effects of the factors identified in qualitative research conducted by Otomo et al. (2023) and investigate whether the factors previously identified as promoting internationalization in small domestic markets also apply to countries with larger domestic markets.

In the next section, we offer 12 hypotheses which are generated by prior research. We then summarize the research methods used in this study. Specifically, quantitative data were collected from 71 startups and analyzed by PLS-SEM. We then assess these hypotheses. Finally, we report on empirical findings and provide practical Implication.

## 2. Research Model and Theoretical Foundations

*2.1. A Multidimensional Perspective on Internationalization Speed*

Drawing upon prior research conducted in small domestic markets as well as the forthcoming study by Otomo et al. (2023), hypotheses were formulated on factors that impact the early internationalization by high-tech startups in large domestic markets.

Some scholars have conceptualized the notion of internationalization in terms of its speed, mainly referring to the time elapsed between a firm's inception and its first international sales (Kiss and Danis 2008; Li et al. 2015). Hsieh et al. (2019) provided empirical evidence to add a multidimensional perspective on (a) the earliness of internationalization, (b) speed of deepening, and (c) speed of geographic diversification as three different

strategic alternatives. Lopez et al. (2009) focus on distinguishing between firms which operate globally and those that exclusively export to neighboring markets, highlighting how viewing the issue from a singular perspective can obscure important differences among firms. Therefore, this study employs a multidimensional approach to examine the speed of internationalization.

Oviatt and McDougall (2005) differentiated three dimensions of internationalization speed as: (1) time between the discovery of an opportunity and the first foreign market entry, (2) how rapidly foreign market entries proceed and how rapidly psychically distant markets are entered, and (3) how quickly international commitments are made, and how fast the percentage of international sales increases. There is an ongoing debate surrounding the notion of internationalization speed due to the varying terminologies (such as pace, rhythm, precocity, early, rapid, accelerated, and time to internationalization) that have been employed in the prior research (Chetty et al. 2014).

The focus of this study is an examination of the three dimensions of internationalization speed. First, we investigate "opportunity" as the speed at which a firm makes an initial entry into the global market. Otomo et al. (2023) differentiate the firms that operate overseas shortly after their inceptions. Second, following the definitions used in many other BGs studies including Oviatt and McDougall (2005), we investigate "speed" as the time between identifying an opportunity and making an initial entry into the foreign market. Third, we look at the "degree" of internationalization beyond the percentage of sales internationalization, as suggested in the study by Lopez et al. (2009), as well as the stage of development and the degree of activities in foreign markets shown by Dunning and Lundan (1993). We further hypothesized the factors that may positively influence the (a) opportunity, (b) speed, and (c) degree of early internationalization.

### 2.2. Hypotheses and Factors That Derive Early Internationalization

One of the determinants influencing a firm's early internationalization is its entrepreneurial characteristics. Many scholars contend that a firm is an extension of an entrepreneur (Gilbert et al. 2006). Bloodgood et al. (1995) found that the founder's direct involvement in international exposure, as well as their experience working for multinational firms, had a direct influence on a company's decision to pursue early internationalization. Therefore, many studies have employed international experience as a proxy for the knowledge and network that allow a firm to pursue a global venture (Fernhaber et al. 2009). A recent study by Paul and Rosado-Serrano (2019) also shows that some specific knowledge and experience are fundamental for early internationalization and performance, such as prior experience or knowledge in related industries (Jiang et al. 2020). Oviatt and McDougall (2005) maintain that entrepreneurs are likely to utilize established networks to explore international opportunities efficiently. Such networks can be a source of information for firms to learn about foreign markets and help entrepreneurs create strategic alliances or partnerships to enhance credibility in a new and unfamiliar market.

Oviatt and McDougall (2005) integrated traditional entrepreneurship literature and international business research to propose the concept of international entrepreneurship and argued that it influences the speed of internationalization. The authors define international entrepreneurship as the discovery, enactment, evaluation, and exploitation of opportunities across national borders to create future goods and services and present a model in which the speed of internationalization is explained by an entrepreneurial opportunity realized by technological innovation and motivated by market competition. Entrepreneurial characteristics also include international vision, which is the ability of an entrepreneur to look beyond domestic markets and anticipate complex connections between design, production, and distribution across international boundaries (Karra et al. 2008). Entrepreneurs are more likely to view the global market as a whole. These authors argue by showing that the early stage of INV's creation appears to be underpinned to a large extent by entrepreneurial characteristics.

Knight and Cavusgil (2004) showed international entrepreneurial orientation plays a critical role in entering the global market early. They showed that International entrepreneurial orientation reflects the firm's overall innovativeness and proactiveness in the pursuit of international markets. It is associated with innovativeness, managerial vision, and proactive competitive posture. They define international entrepreneurial orientation based on quantitative measurements. Otomo et al. (2023) also suggests that international entrepreneurial orientation influences firm's early internationalization in firms founded in countries with a large domestic market.

Based on the above literature, we generated the following hypothesis:

**H1:** *Entrepreneurs' international orientation has a positive effect on firm's (a) opportunity, (b) speed, and (c) degree of early internationalization.*

Knowledge of foreign languages helps nurture a global mindset and thus serves as a prerequisite for aspiring global entrepreneurs to succeed (Cannone and Ughetto 2014). Otomo et al. (2023) found that an entrepreneur's communication skills, in addition to the knowledge of foreign languages, promote successful early internationalization. We thus propose the following hypotheses:

**H2:** *Entrepreneur's knowledge of foreign languages has a positive effect on firm's (a) opportunity (b) speed, and (c) degree of early internationalization.*

**H3:** *Entrepreneur's communication skill has a positive effect on firm's (a) opportunity, (b) speed, and (c) degree of early internationalization.*

The resource-based view (Wernerfelt 1984) suggest that the internal factors influence a firm's early internationalizing firms and their performance. A firm's performance in global markets is impacted by the factors other than individual traits of entrepreneurs. Knight and Cavusgil (2004) found that factors such as the distinctiveness of technology and innovative culture affect a firm's international performance. The scalability of products also affects a firm's decision for early internationalization; computer hardware as well as software applications are examples of such scalable products (Cannone and Ughetto 2014).

According to Knight and Cavusgil (2005), firms that engage in early internationalization tend to differentiate their focus by targeting niche markets and foregoing cost leadership. This is because a cost leadership strategy will expose firms to intense price competition and brings little advantage for small firms that are considering early internationalization. Instead, as a key component of a product differentiation strategy, firms rely on their technological expertise (Lisboa et al. 2011), and new and small ventures with a high R&D intensity are prone to initiate internationalization within three years of their inception (Li et al. 2015). These findings were confirmed by Otomo et al. (2023).

Thus, based on the literature cited above, we have formulated the following hypotheses:

**H4:** *Distinctiveness of a firm's technology has a positive effect on (a) opportunity, (b) speed, and (c) degree of early internationalization.*

**H5:** *Scalability of products has a positive effect on firm's (a) opportunity, (b) speed, and (c) degree of early internationalization.*

**H6:** *A firm's R&D intensity has a positive effect on (a) opportunity, (b) speed, and (c) degree of early internationalization.*

Mudambi and Zahra (2007) highlighted the importance of the international experience within a firm's top management team. They emphasized that entrepreneurs should give particular attention to and make the most of the top management members' international experience. Otomo et al. (2023) found that, in small firms, global experiences of the members beyond the top management also influence a firm's decision to internationalize early.

**H7:** *The international exposure of a firm's members has a positive effect on firm's (a) opportunity, (b) speed, and (c) degree of early internationalization.*

Shibahara (2017) focused on several firms which had initially targeted the domestic market but quicky pursued internationalization and identified several factors that led to their success. He concluded that these firms relied on an external support structure which provided information relevant to their product, introduced them to the local networks, as well as mental support and extra trainings. The interview findings in Otomo et al. (2023) highlight a case where firms achieve internationalization by obtaining support from public organizations.

**H8:** *Obtaining external support has a positive effect on firm's (a) opportunity, (b) speed, and (c) degree of early internationalization.*

In addition, environmental factors also affect firm's early internationalization. Fernhaber et al. (2007) investigated how environmental factors, such as industrial structure, affect the extent of internationalization, and found that an industry with a higher level of knowledge intensity is more likely to pursue internationalization. Further, a company's likelihood of pursuing early internationalization is enhanced if other local industries are globally interconnected. The diverse set of factors collectively determine a firm's propensity toward early internationalization. Deregulation also promotes early internationalization (Chetty and Campbell-Hunt 2004). Otomo et al. (2023) also found that conditions surrounding the domestic market, including regulatory environment and culture that may discourage entrepreneurial efforts, can also serve as a driving force behind a firm's decision to pursue international operation early. We thus generated the following hypothesis:

**H9:** *Conditions surrounding the domestic market have a positive effect on firm's (a) opportunity, (b) speed, and (c) degree of early internationalization.*

Through interviews, Otomo et al. (2023) identified three new factors not previously introduced in the literature: global needs, geographic considerations, and types of technology. Global needs refer to basic needs that have broad appeal. When firms have a core technology that has an universal appeal across different cultures and markets, they have are likely to have an embedded intention to go global from their inception. In contrast, geographic considerations, including the physical distance between a firm's home country and the targeted foreign market, as well as the type of technology, measured by the engineering complexity and the length of market penetration, were identified and discussed as potential barriers to early internationalization. (see Figure 1)

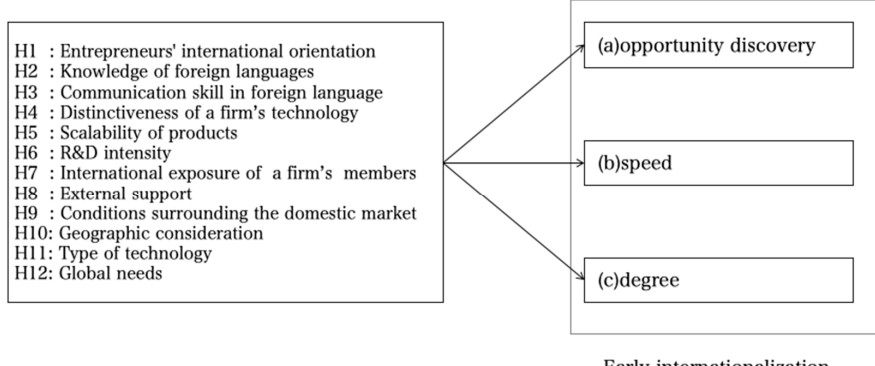

**Figure 1.** Conceptual framework.

**H10:** *Geographic consideration has a positive effect on a firm's (a) opportunity, (b) speed, and (c) degree of early internationalization.*

**H11:** *Type of technology has a positive effect on a firm's (a) opportunity, (b) speed, and (c) degree of early internationalization.*

**H12:** *Global needs has a positive effect on a firm's (a) opportunity, (b) speed, and (c) degree of early internationalization.*

## 3. Research Methodology

### 3.1. Data Collection and Sample

The research data were collected among the startups in Japan. The theoretical reasons for our case selection are twofold.

First, Japan has a large enough domestic market, which suggests limited incentive for internationalization, and thus poses a puzzle. The World Bank database shows Japan's gross national income (GNI) in 2021 as ranking third in the world. Second, Japan is characterized by a significant psychological distance from its neighboring countries, both linguistically and geographically. In the traditional internationalization model (Johanson and Vahlne 1977), psychological distance to the target county is considered an important determinant for a firm's decision to pursue early internationalization. Taking into account the potential risk that might be associated with psychological distance, entrepreneurs in Japan are likely to have less incentive to internationalize compared to those in countries with larger domestic markets, such as the United States. As a result, Japanese small firms seeking early internationalization pose a puzzle to the existing literature.

An online questionnaire was distributed and collected from firms via Microsoft Forms. A total of 404 firms received messages including the questionnaire link along with the objective of the study and the survey instructions. In addition to the respondents to the qualitative survey conducted by Otomo et al. (2023), firms that had participated in overseas exhibitions and those selected for J-Startup, a startup incubation program by the Ministry of Economy, Trade, and Industry, received messages. A total of 71 responses were collected. Similar to the previous study (Otomo et al. 2023), the questionnaire was specifically designed for individuals who have the authority to oversee international operations and expansion efforts.

Table 1 lists a summary of the respondents' characteristics. The majority (52%) were small firms with fewer than 10 employees, while the information and telecommunications, and manufacturing industries combined accounted for 75%. On average, the respondents reported JPY 343 million in annual sales for the most recent fiscal year (76% are unrevealed annual sales). Many of the responding firms were small, technology-focused startups.

**Table 1.** Characteristics of the respondents.

| Characteristics | N = 71 |
|---|---|
| a. Number of employees | |
| 10–1 月 | 37(52%) |
| 11–50 | 23(32%) |
| 51–100 | 7(10%) |
| 101–300 | 3(4%) |
| 301– | 1(1%) |
| b. Types of industry | |
| Information & Communication | 29(41%) |
| Manufacturing | |
| Medical Welfare | 24(34%) |
| Services | 9(13%) |
| Agriculture, forestry, fisheries, steel | 4(6%) |
| Education, Learning support | 3(4%) |
| Retail | |
| c. Types of the respondent | 1(1%) |
| CEO | |
| CXO | 1(1%) |
| others | |
| | 52(73%) |
| | 7(10%) |
| | 12(17%) |

### 3.2. Instrument Development

A questionnaire consisting of 20 questions, written in the native language of the respondents, Japanese, was used for data collection: 7 questions were asked about the demographics, 4 questions for deal with this survey's results, and 5 questions were used to measure their propensity for early internationalization. Then, the degree of each firm's internationalization was assessed based on a score between 1 and 6 by applying Dunning and Lundan's (1993) developmental stage theory. To evaluate the opportunity and speed of a firm's entry into a foreign market, we used a scale that ranged from "less than 1 year," to "within 3 years, 3–10 years," and "more than 10 years".

The remaining four questions were used to measure the following variables included in our hypotheses: the entrepreneur (10 items), the firm (15 items), outside experts (4 items), and patents (1 item). We used a five-point Likert scale ranging from (1) not at all, to (2) disagree, slightly disagree, (3) undecided, (4) slightly agree, and (5) strongly agree.

The questions about entrepreneurs were used to test three hypotheses: H1, H2, and H3. Further, H1 assessed entrepreneurs' international orientation using the seven questions from a previous study by Knight and Cavusgil (2004). H2 was based on a definition by Cannone and Ughetto (2014), and H8 was based on a study by Shibahara (2017). H10–H13 were identified from qualitative research conducted by Otomo et al. (2023), and the questions to test these hypotheses were generated based on the interview's script. In addition, to ensure that the questions were easy to understand and answer, a pilot study was conducted in advance.

### 3.3. Data Analysis

The SEM-PLS approach based on the SmartPLS version of 4.0.8.3. PLS-SEM was used for the data analysis; PLS-SEM aims at maximizing the R-square values in the path model and offers a flexible and powerful approach to multivariate analysis with small samples (Smith and Barclay 1997). Since this study develops measurement models consisting of formative indicators, the assessment is performed through different reliability and validity tests following the protocols used in Hair et al. (2017).

First, to assess collinearity issues, by following in Hair et al. (2017), we looked at the Variance Inflation Factor (VIF) value. If the level of collinearity was very high and the VIF value indicated 5 or higher, we considered removing one of the corresponding indicators. We then examined the outer weight and outer loading of each indicator by using 5000 bootstrapping in order to assess their significance. If an indicator did not prove to be significant, we removed it from the formative model.

As for the evaluation of the structural model, the VIF, R-square value, and F-square were examined by using bootstrapping with 5000 interactions. These analyses were conducted at a 0.10 level of significance. R-square statistics explain the variance in the endogenous variable explained by the exogenous variables. Following the model presented by Hair et al. (2017), R-square values of 0.75, 0.5, and 0.25 were used to represent substantial, moderate, and weak relationships, respectively. In a structural mode, F-square means that a variable may be affected by several different variables. Removing an exogenous variable can affect the dependent variable. Assessing F-square is that values of 0.02, 0.15, and 0.35 represent small, medium, and large effects, respectively (Cohen 1988). Effect size values less than 0.02 indicate that there is no effect.

To test mediating effects and the hypothesized relationships, mediation analysis was conducted following the model presented by Hair et al. (2017). To test the type of mediation, the significance of the indirect effect and/or direct effect were assessed.

## 4. Result

### 4.1. Measurement Model

Regarding collinearity, all VIF value indicators were less than five and thus proven acceptable. Similarly, their significance was evaluated by using the model in Hair et al. (2017),

and ten indicators were removed as a result. Table 2 lists the detailed results after the removal of these indicators.

**Table 2.** Assess Measurement Model.

| | *p* Value (Outer Weight) | Outer Loadings | *p* Value (Outer Loading) |
|---|---|---|---|
| Type of technology | 0 | 1 | 0 |
| Domestic market condition 1 | 0.889 | 0.417 | 0.117 |
| Domestic market condition 2 | 0.028 | 0.779 | 0 |
| Domestic market condition 3 | 0.012 | 0.795 | 0 |
| Entrepreneur's International orientation 1 | 0.005 | 0.878 | 0 |
| Entrepreneur's International orientation 2 | 0.928 | 0.678 | 0.001 |
| Entrepreneur's International orientation 3 | 0.248 | 0.673 | 0.001 |
| Entrepreneur's International orientation 5 | 0.448 | 0.486 | 0.02 |
| Entrepreneur's International orientation 6 | 0.663 | 0.428 | 0.076 |
| Entrepreneur's International orientation 7 | 0.34 | 0.392 | 0.106 |
| Communication skills | 0 | 1 | 0 |
| Knowledge of foreign language | 0 | 1 | 0 |
| External support 2 | 0.12 | 0.97 | 0.012 |
| External support 4 | 0.653 | 0.614 | 0.16 |
| Distinctiveness of a firm's technology 1 | 0.322 | −0.634 | 0.329 |
| Distinctiveness of a firm's technology 2 | 0.264 | 0.722 | 0.265 |
| Scalability 1 | 0.546 | 0.74 | 0.042 |
| Scalability 2 | 0.167 | 0.952 | 0.005 |
| Global needs | 0 | 1 | 0 |
| Geographic consideration 1 | 0.126 | 0.966 | 0 |
| Geographic consideration 2 | 0.492 | 0.851 | 0 |
| The international exposure of a firm's members | 0 | 1 | 0 |
| R&D intensity | 0 | 1 | 0 |

### 4.2. Structural Model

The evaluation of the structural model indicates that the hypotheses below were supported. This was based on the T-values greater than 1.65, the *p*-value being less than 0.10, and the bootstrap confidence interval bias not crossing a 0 mark between the upper and lower intervals. Hypotheses H12-(a), H3-(a), H12-(b), H11-(c), and H10-(c) were found to be supported at a significance level of 0.10, while hypothesis H2-(a) was supported at a significant level of 0.05 level. Hypotheses H1-(b) and H9-(a) were supported at a significance level of 0.01.

The F-square values, which were all between 0.02 and 0.15, suggest a small effect size for the variables. The opportunity, speed, and degree had R-square values of 0.312, 0.28, and 0.328, respectively, indicating relatively moderate strength compared against other variables. The specific values are listed in Table 3.

We conducted a mediation analysis and found that the mediated path between speed of internationalization from the domestic market conditions via an entrepreneur's international orientation is significantly and positively supported (Table 4). According to Hair et al. (2017), the indirect effect and the direct effect are both significant and point in the same direction, so the type of mediation was a complementary partial mediation.

**Table 3.** Direct Effects.

| | Hypotheses | Original Sample (O) | T Statistics | *p* | VIF | R^2 | f^2 |
|---|---|---|---|---|---|---|---|
| **(a) Opportunity discovery** | Domestic market condition | 0.135 | 2.866 | *** | 1.527 | 0.328 | 0.044 |
| | Global needs | 0.117 | 1.794 | * | 1.185 | | 0.054 |
| | Knowledge of foreign language | 0.164 | 2.28 | ** | 1.938 | 0.28 | 0.137 |
| | Communication skills | 0.179 | 1.715 | * | 2.58 | | 0.06 |
| **(b) Speed** | Entrepreneur's International orientation | 0.15 | 2.832 | *** | 1.839 | | 0.143 |
| | Global needs | 0.119 | 1.895 | * | 1.185 | 0.312 | 0.054 |
| **(c) Degree** | Type of technology | 0.103 | 1.763 | * | 1.114 | | 0.105 |
| | Geographic consideration | 0.123 | 1.776 | * | 1.31 | | 0.053 |

\* $p < 0.1$; \*\* $p < 0.05$; \*\*\* $p < 0.001$.

**Table 4.** Mediation (Indirect) Effects.

| | Original Sample (O) | T Statistics | *p* |
|---|---|---|---|
| Domestic market condition → Entrepreneur's International orientation → Speed | 0.099 | 1.741 | * |

\* $p < 0.1$; \*\* $p < 0.05$; \*\*\* $p < 0.001$.

## 5. Research Findings

Our study focused on Japanese high-tech startups to investigate how firms founded in a country with a large domestic market would achieve early internationalization despite the absence of strong incentives to operate overseas. The hypotheses were derived from the qualitative analysis of Otomo et al. (2023) and were tested using the partial least squares structural equation modeling with Smart PLS. The findings revealed that there were eight direct paths and one mediated path that showed statistically significant.

Entrepreneur's international orientation (H1) was found to be a crucial factor in explaining a firm's early internationalization, even when the originating market size is significant. This result was consistent with the previous research findings that highlighted the influence of entrepreneurial characteristics on a firm's decision to internationalize (Gilbert et al. 2006; Bloodgood et al. 1995; Otomo et al. 2023). This study suggests that the international orientation of an entrepreneur can drive a firm to pursue global business at an early stage, regardless of the size of the domestic market.

Our results also showed that an entrepreneur's knowledge of foreign language (H2) and communication skills (H3) had a positive impact on a firm's ability to cultivate opportunities for early internationalization at a significant level of 0.05 level and 0.10 level, respectively.

Japan is characterized by a significant psychological distance from its neighboring countries, both linguistically and geographically. Entrepreneurs' familiarity with the language spoken in a targeted market as well as effective communication skills can help them become more aware of new opportunities to enter a new market. This aligns with the findings of Cannone and Ughetto (2014), which showed that knowledge of a foreign language helps to develop an international mindset.

Otomo et al. (2023) found that, in some cases, the conditions in the domestic market can drive firms to internationalize despite the lack of an inherent motivation to operate overseas (H9). Factors such as insufficient market size, legal and policy constraints, and unfavorable business practices might drive startup firms out of a domestic market.

The path coefficient of 1% between domestic market conditions and the identification of internationalization opportunities is significant, indicating a clear influence of these conditions on a startup firm's decision to pursue internationalization at an early stage. This finding aligns with Kudina et al. (2008), who reported that, in the UK, the firms were

forced to operate internationally due to the scarce demand for their particular products even though the domestic market as a whole was large. These results show that factors promoting a firm's internationalization would also apply in countries with large domestic markets, contributing to the ongoing discussion of the internationalization processes.

On the other hand, no statistical significance was found for the factors such as "Distinctiveness of a firm's technology" (H4), "Scalability of products" (H5), "R&D intensity" (H6), and "External support" (H8).

This could have been a result of a possible lack of clarity and shortcomings in some of the survey questions. One pilot study was not enough to ensure that the questions

As identified in Otomo et al. (2023) as a factor affecting a firm's decision to internationalize, our study showed a significant 10% path coefficient of "geographical consideration" (H10), indicating that it could significantly influence the degree of internationalization; a degree of a firm's penetration into an overseas markets could be much deeper when there is less physical distance between the originating country and the targeted country.

Some scholars on born global studies have decided to drop geographic consideration from the discussion, pointing to the advancements in information technology (Knight and Cavusgil 2004). However, Otomo et al. (2023) argue that geographic consideration remains a relevant factor in early internationalization. Our findings support this argument, and thus we would like to emphasize the importance of taking physical distance in our future discussions on born global firms.

Lastly, we tested to see if the type of technology plays a significant in a firm's decision to pursue an early internationalization, as indicated by Otomo et al. (2023). Our finding suggested that the rate and speed with which a firm enters a foreign market varied significantly even among tech-oriented firms; the more complex the technology is, the longer it took for a startup to operate overseas. The path coefficient to the degree of internationalization was significant at 10%.

## 6. Conclusions

Early internationalization and its success in foreign markets have a significant impact on both a firm's growth and the global economy. While once seen as a practice observed only among limited countries with small domestic markets is now common throughout the world. Early internationalization has thus become a crucial strategy for all firms, even when operating in a large domestic market. To test what factors might drive high-tech startups operating in a large domestic market into early internationalization, this study developed hypotheses based on 13 factors identified in the qualitative investigation of Otomo et al. (2023) as positively influencing the (a) opportunity, (b) speed, and (c) degree of early internationalization.

Data from 71 firms were collected through an online questionnaire survey and were analyzed using Structural Equation Modeling (SEM). PLS-SEM is well-suited for small sample sizes; we utilized a formative model. The study found a correlation between early internationalization and several factors, including entrepreneurs' international orientation, domestic market condition, global needs, the entrepreneur's communication skills and familiarity with the language spoken in the target market, geographic considerations, and type of technology that the firm is pursuing. Our study demonstrates that many of the factors identified in previous research as promoting internationalization in countries with small domestic markets would apply in countries with large domestic markets. Therefore, it makes a theoretical contribution by expanding the existing discussion on internationalization processes previously introduced by Kudina et al. (2008).

The study found positive evidence for the influence that the type of technology, geographic consideration, and global needs may have on early internationalization. It highlighted the importance of geographic factors in discussing and understanding a firm's decision to enter a foreign market shortly after its inception, a point which was previously overlooked in the literature on early internationalization.

Furthermore, the mediator analysis suggests that certain characteristics of a domestic market might provide disincentives for startups to remain in the country and accelerate the speed of internationalization if the entrepreneur possesses international orientation. Thus, policymakers seeking to promote entrepreneurship and enhance international competitiveness of domestic firms through early internationalization must carefully balance these factors.

### 6.1. Practical Implication

This study, which tested the factors that impact early internationalization, provides practical suggestions for firms aiming to pursue early internationalization.

The global needs (H12) were found to have a significant effect on a firm's (a) opportunity and (b) speed of early internationalization. While a niche strategy is widely regarded as being crucial to achieve success in early internationalization (Knight and Cavusgil 2005), firms may also succeed by identifying and focusing on problems that people in different parts of the world commonly consider being challenging.

Our study supports the findings of Otomo et al. (2023) that the types of technology can impact a firm's internationalization effort. The more straightforward the technology, the easier it is to penetrate markets overseas. The novelty of a technology can serve as a firm's competitive advantage, however, it can also hinder its ability to successfully expand overseas. For startups with internationalization as their ultimate goal, it may be more advisable to focus on focusing on simple technology as its core commodity.

Innovation can be divided into two stages: product innovation, which involves the introduction of new products, and process innovation, which improves firm productivity (Kaneko 2017). Firms handling complex technology may be able to compete through product innovation; however, overcoming the "liability of newness" (Stinchcome 1965) or determining a dominant design can be time-consuming, making it challenging to achieve process innovation and eventually penetrate foreign markets.

Kraus et al. (2017) found that the business model of Born Globals tends to be more efficiency-centered rather than novelty-centered. In terms of technological complexity, firms may benefit from choosing simple, efficient technologies that allow for faster market penetration, rather than overly complex, novel technologies.

Based on an investigation on Finnish BGs, Luostarinen and Gabrielsson (2006) provided policy implications for politicians by highlighting challenges and presenting solutions. Similarly, the findings of our study provide practical implications for policymakers who aim to promote early internationalization efforts.

### 6.2. Limitations and Direction for Future Research

In conclusion, we would like to briefly point out a limitation of our study and suggest potential areas for future research.

The majority of the samples in our study sample were taken from the list of firms provided by a semi-public organization that supports startups in Japan. As a result, the responses to questions about the impact of public institutions on early internationalization may have been influenced by a bias in the sample collection.

Further, expanding the investigations both geographically and longitudinally would be crucial in furthering our understanding of this subject. In order to eliminate country-specific noises, case studies need to be conducted in countries with a large domestic market. This will offer more data points and enhance our understanding of the specific mechanism involved in a firm's globalization process. In addition, longitudinal studies into the future performance will add more insights and improve our understanding of a firm's internationalization efforts.

**Author Contributions:** Conceptualization, S.O.; methodology, S.O.; software, S.O.; validation, S.O.; formal analysis, S.O.; resources, S.O.; writing—original draft preparation, S.O.; writing—review and editing, S.I. and M.Y.-Y.; supervision, S.I. and M.Y.-Y.; project administration, S.I. All authors have read and agreed to the published version of the manuscript.

**Funding:** This research was funded by JPPS KAKENHI Grant Numbers JP19H01516, JP20H01546, and JP22K01742.

**Institutional Review Board Statement:** Not applicable.

**Informed Consent Statement:** Not applicable.

**Data Availability Statement:** The data presented in this study are available on request from the corresponding author.

**Acknowledgments:** The authors would like to appreciate all the companies that participated in the survey and provided valuable insights into our research, with special appreciation to Tenchijin Inc. for their cooperation, which provided an internship opportunity at their organization.

**Conflicts of Interest:** The authors declare no conflict of interest.

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
