# Peer review of "Empirical Research on Early Internationalization of Firms in Sufficiently-Sized Domestic Market Country"

_admsci, doi:10.3390/admsci13040107_

Round 1

Reviewer 1 Report

In the introduction section, I was expecting the motives of internationalization, such as one of the strategies for growth. Many firms fail at the early stage while these firms are growing internationally. There must have some motivations behind this, and you need to clarify this in the introduction. Why have you applied the Uppsala model instead of other internationalization theories or even a resource-based view? A better justification is required.

In the third paragraph, you mentioned that- ‘This phenomenon has been most prevalent among knowledge-intensive firms including ones that specialize in software or information technology products that operate in small countries because the small domestic market drives them to seek opportunities beyond national borders and facilitate their entry into the international market’. Advantages in technologies and positive changes in socio-economic settings are also influencing this. You can just touch several drivers of internationalization here with one reference for each driver.

Your discussion on the data collection from 13 high-tech Japanese firms is more suitable for the methodology section than the introduction. Instead, you can briefly state- how the rest of the sections are organized in this paper.

Most of the references in section 2.2 are not recent enough to justify the relevance now. While you can keep some of these old references, you should also use some recent references.

Some of the item loadings are lower than usual and you need to write a sentence to justify this. You should also add a couple of sentences to highlight the techniques that you have used to tackle nonresponse and early response biases. 

Author Response

Thank you for providing these insights
All comments are answered below and in the attached. 

In the introduction section, I was expecting the motives of internationalization, such as one of the strategies for growth. Many firms fail at the early stage while these firms are growing internationally. There must have some motivations behind this, and you need to clarify this in the introduction.

→Purpose of this paper is to find what factors that derive early internationalization. The future performance(fail/success) would be considered on my next research.

Why have you applied the Uppsala model instead of other internationalization theories or even a resource-based view? A better justification is required.

→ Traditional internationalization approach was incremental approach and Uppsala model was one of the most representative theory. We consider multiple theories and added the sentence related resource-based view on 2.2

In the third paragraph, you mentioned that- ‘This phenomenon has been most prevalent among knowledge-intensive firms including ones that specialize in software or information technology products that operate in small countries because the small domestic market drives them to seek opportunities beyond national borders and facilitate their entry into the international market’. Advantages in technologies and positive changes in socio-economic settings are also influencing this. You can just touch several drivers of internationalization here with one reference for each driver.

→ Thank you for efficient advice. I had touched many references to emphasis previous research focus the phenomen on small country.

Your discussion on the data collection from 13 high-tech Japanese firms is more suitable for the methodology section than the introduction. Instead, you can briefly state- how the rest of the sections are organized in this paper.

→ I have added section about how the rest of the sections are organized on the final part of introduction. “In the next section, we offer 12 hypotheses which generated by prior research. We then summarize the research methods used in this study. Specifically, quantitative data were collected from 71 startups and analyzed by PLS-SEM. We then assess these hypotheses. Finally, we report on empirical findings and provide practical Implication.

Most of the references in section 2.2 are not recent enough to justify the relevance now. While you can keep some of these old references, you should also use some recent references.

→ Jiang et al., (2020) and Paul & Rosado-Serrano (2019) is added as reference

Some of the item loadings are lower than usual and you need to write a sentence to justify this. You should also add a couple of sentences to highlight the techniques that you have used to tackle nonresponse and early response biases. 

→ Item loadings validation are following PLS-SEM model book,  Hair et al. (2017).

Reviewer 2 Report

The purpose of the research is to investigate the factors that influence the high-tech startups in a large domestic market to achieve early internationalization, despite the absence of strong incentive to operate overseas. The topic is interesting, however, the paper needs to be improved methodologically and in exposition.

-          Please explain the meaning of PLS-SEM in abstract.

-          The motivation and the contributions of the paper need to be written in Introduction.

-          At the end of Introduction section, please mention the structure of the paper.

-          I appreciate the authors’ effort to formulate the hypothesis, however, the hypothesis need to be better documented. The authors have to improve both theoretical discussions to support their expectations as well as the empirical design to test their hypotheses. For the first issue, the literature review has to be extended with clear discussions supporting the factors that influence early internationalization.

-          The methodology should be improved. I don’t see any formula used to perform the research. Please document the model used. Also, please explain the selection of the variables.

-          A table for descriptive statistics by country will give a better view for the sample composition.

Author Response

Thank you for providing these insights
All comments are answered below and in the attached. 

  • Please explain the meaning of PLS-SEM in abstract.

→ “PLS-SEM(Partial least squares path modeling)” is added on abstract

  • The motivation and the contributions of the paper need to be written in Introduction.

→ I added the sentence at the end of introduction “Our study will make  contributions at multiple levels, benefiting entrepreneurs who are contemplating y international operations  in the future as well as policymakers who aim to promote early internationalization efforts.”

  • At the end of Introduction section, please mention the structure of the paper.

The sentences “In the next section, we offer 12 hypotheses which generated by prior research. We then summarize the research methods used in this study. Specifically, quantitative data were collected from 71 startups and analyzed by PLS-SEM. We then assess these hypotheses. Finally, we report on empirical findings and provide practical Implication.” were added .

  • I appreciate the authors’ effort to formulate the hypothesis, however, the hypothesis need to be better documented. The authors have to improve both theoretical discussions to support their expectations as well as the empirical design to test their hypotheses. For the first issue, the literature review has to be extended with clear discussions supporting the factors that influence early internationalization.

→ I have added some references, one is about resource-based view and other are insights from recent papers ( Jiang et al., (2020) and Paul & Rosado-Serrano (2019)) to extend and improve theoretical discussion.

The number of papers which conducted study in same environment with me which was in Japan was very limited and it is difficult to confirm the empirical design to test their hypotheses for now.

  • The methodology should be improved. I don’t see any formula used to perform the research. Please document the model used. Also, please explain the selection of the variables.

→formula are documented on one of the reference Hair et al. (2017) which was big book of PLS-SEM model. Selection of the variables are also following Hair et al. (2017)’s method and added sentence on 3.3

  • A table for descriptive statistics by country will give a better view for the sample composition.

→In this study, all sample are from Japan. When cross-country survey as my next research conduct, I will use a table for descriptive statistics by country.

Round 2

Reviewer 1 Report

Thank you for sending the paper again to review. I have carefully read the manuscript in addition to the 'response letter'. The logical flow is clear now. I can also see that the contribution has improved substantially. I enjoyed reading the article. The author (s) have/has addressed all the concerns raised in my review. Therefore, I am happy with its current form. Just a minor correction related to the referencing style as required by the MDPI journals. 

Author Response

Thank you for sending the review again. I'm glad to hear your comment. About  the referencing style, I think editing team will revise before publishing and authors may not need to care. 

Thank you

Reviewer 2 Report

The authors should work more on explaining the contribution of the paper on both theoretical and empirical levels. It is not enough to mention "Our study will make contributions at multiple levels.."  You should clarify the contributions of the paper which are not elaborated well in the current paper. You can talk about the following contributions: What insights can you provide based on your finding? Do they push forward our understanding? What should we do with your research? Do you have any suggestions to improve the current regulation or practice?

Author Response

Thank you for providing these insights

I worked more on explaining the contribution of the paper on both theoretical and empirical levels.

First, I added brief explanation about the contributions on the end of introduction part.

“Our study will make  contributions at multiple levels, benefiting entrepreneurs who are contemplating  international operations  in the future by helping to generate hypotheses that can serve as guiding posts for firms considering overseas operations as well as policymakers who aim to promote early internationalization efforts by providing insight what factor external factors are asked to promote firms international business.

Theoretically, our study will expand the existing discussion on internationalization processes. Specifically, we will examine the effects of the factors identified in a qualitative research conducted by Otomo et al. (2023, forthcoming), and investigate whether the factors previously identified as promoting internationalization in small domestic markets also apply to countries with larger domestic markets.”

Detailed explanation of contribution are written on conclusion part.

“This study found a correlation between early internationalization and several factors, including entrepreneurs' international orientation, domestic market condition, global needs, the entrepreneur’s communication skills and familiarity with the language spoken in the target market, geographic considerations, and type of technology that the firm is pursuing. Our study demonstrates that many of the factors identified in previous research as promoting internationalization in countries with small domestic markets, would apply in countries with large domestic markets. Therefore, it makes a theoretical contribution  by expanding  the existing discussion on internationalization processes previously introduced by  Kudina, Yip, and Barkema (2008).”

The study found positive evidence for the influence that the type of technology, geographic consideration, and global needs may have on early internationalization. It highlighted the importance of geographic factors in discussing and understanding  a firm’s decision to enter a foreign market shortly after its inception, a point which was previously overlooked in the literature on early internationalization.

Furthermore, the mediator analysis suggests that certain characteristics of a domestic market might provide disincentives for startups to remain in the country and accelerate the speed of internationalization. if  the entrepreneur possesses  international orientation. Thus, policymakers seeking to promote entrepreneurship and enhance international competitiveness of domestic firms through early internationalization must carefully balance these factors.”